# Survival of Self-Replicating Molecules under Transient Compartmentalization with Natural Selection

**DOI:** 10.3390/life9040078

**Published:** 2019-10-03

**Authors:** Gabin Laurent, Luca Peliti, David Lacoste

**Affiliations:** 1UMR CNRS Gulliver 7083, ESPCI, 10 rue Vauquelin, CEDEX 05, 75231 Paris, France; gabin.laurent@u-psud.fr; 2Santa Marinella Research Institute, 00052 Santa Marinella, Italy; luca@peliti.org

**Keywords:** origin of life, error catastrophe, parasites

## Abstract

The problem of the emergence and survival of self-replicating molecules in origin-of-life scenarios is plagued by the error catastrophe, which is usually escaped by considering effects of compartmentalization, as in the stochastic corrector model. By addressing the problem in a simple system composed of a self-replicating molecule (a replicase) and a parasite molecule that needs the replicase for copying itself, we show that transient (rather than permanent) compartmentalization is sufficient to the task. We also exhibit a regime in which the concentrations of the two kinds of molecules undergo sustained oscillations. Our model should be relevant not only for origin-of-life scenarios but also for describing directed evolution experiments, which increasingly rely on transient compartmentalization with pooling and natural selection.

## 1. Introduction

Research on the origins of life is plagued by several chicken-and-egg problems [1]. One central problem concerns the emergence of functional self-replicating molecules. To be a functional replicator, a molecule must be long enough to carry sufficient information, but if it is too long it cannot be replicated accurately, because shorter non-functional molecules called parasites may replicate faster and take over the system. This was experimentally observed many years ago by Spiegelman [2]. This observation was then rationalized using the notion of error threshold [3], which plays a key role in research on the origins of life [4].

Several theoretical solutions have been proposed to address this issue, among which the stochastic corrector model [5,6] is prominent. In this model, small groups of replicating molecules grow in compartments, to a fixed final size called the carrying capacity. Then, the compartments are divided and their contents are stochastically partitioned between the two daughter compartments. Thanks to the variability introduced by this stochastic division, and to the selection acting on the compartments, functional replicators can be maintained in the presence of parasites.

For a long time, these theoretical ideas have lacked an experimental illustration. This changed in 2016 thanks to progresses in the manipulation of in vitro molecular systems. That year using such systems, Matsumura et al. demonstrated that transient compartmentalization is indeed able to maintain RNA replicators despite the presence of RNA parasites [7], as predicted by the Stochastic corrector model. Then, Bansho et al. built an in vitro molecular ecosystem, based on a different compartmentalized RNA, which was able to display host-parasite oscillatory dynamics and evolution [8].

Inspired by the work of S. Matsumura, we recently explored general scenarios of transient compartmentalization that are also able to maintain information in replicating systems [9,10]. We proposed a transient compartmentalization dynamics with no cell division, which should be achievable when only prebiotic chemistry is available. In our framework, there are no specific requirements regarding the chemical composition or the topology of the compartment boundaries: transient compartmentalization can result from environment fluctuations due to day-night cycles [11], tides cycles [12] or cycles of confinement and release of chemicals from pores [13].

In this paper, we extend the framework in [10] to the case of transient compartmentalization of self-replicating molecules. The main new element in this extension is that the selective pressures acting on this system are not externally imposed, as in our previous work, but stem from the system intrinsic dynamics acting as a form of natural selection. Therefore, the assumptions of this new model are agnostic about the environment and its interaction with the system, which is a desirable feature for scenarios on the origins of life. Besides, this extension may be also pertinent for certain in vitro evolution experiments [14]. Indeed, in vitro evolution experiments based on external selection are often more difficult and cumbersome to carry out than those based on natural evolution, which we consider here.

In one version of our model, we find oscillatory behavior in the population size of replicators. These oscillations are different from the ones observed with bulk hypercycles [15], because they only exist due to the transient compartmentalization dynamics. The oscillations present in our model are similar to the ones which have been observed in the molecular ecosystems mentioned above [8]. In the Section 4 we compare the predictions of our model to these experiments and to the theoretical model [16] developed to analyze them. While our framework is applicable to such experimental systems, it is important to appreciate that it has a wide generality. It could equally well describe many other forms of compartmentalized hypercycles or coupled autocatalytic sets, because the self-replicating molecules which we consider need not be RNA replicases.

## 2. Materials and Methods

Here, we introduce two models describing a transient compartmentalization process in which self-replicating molecules (the replicase) may coexist with non-self replicating ones (the parasites) which may be replicated by the first ones. These models are amenable to mathematical analysis. In Section 1, we describe a model of transient compartmentalization where the compartments are populated at each round with an inoculum which has a fixed average size λ as shown in Figure 1a. In Section 2, the size of the inoculum λ(t) is allowed to vary in time as a result of successive dilution steps as shown in Figure 1b.

### 2.1. Transient Compartmentalization with a Fixed Inoculum Size

In this subsection, we describe the behavior of a compartmentalized self-replicating system made of two species: self-replicating molecules (replicases) and parasites. Replicases can make copies of themselves and of other parasite molecules, while the parasites can only be copied by the replicases. This is a different case from that discussed in [10], where both the molecules of interests (in that case, the ribozymes) and the parasites could be replicated by externally provided enzymes. In further contrast, in the present case, there is no externally applied selection. Thus, the main steps of the replicating cycle in this case are as shown in Figure 1a:Inoculate the compartments.Maturate the compartments.Pool compartment contents.

In the inoculation step, as in [10], one chooses a number *n* of molecules from the pool, where *n* is Poisson distributed with average λ. The resulting inoculum then contains *m* replicases and y=n−m parasites, which are distributed according to a Binomial distribution of parameter *x*, where *x* is the initial fraction of replicases in the pool. We denote by $P_{\lambda }\left(n,m,x\right)$ the resulting probability distribution. This follows closely the corresponding steps in [10]. However, the dynamics of the maturation step is different and is described by the following equations:(1)$$\begin{array}{cc} \dot{m}\left(t\right) & =\alpha \;m{\left(t\right)}^{2}\;,\\ \dot{y}\left(t\right) & =\gamma \;m(t)\;y(t)\;, \end{array}$$
where m(t) and y(t) are, respectively, the self-replicating and the parasitic species populations at time *t*, while α and γ are their respective replication rates. The analytical solution described in Appendix A yields the compartment composition (m(T),y(T)) at the stopping time *T* as a function of the initial composition, denoted by (m=m(0),y=y(0)). The stopping time *T* is itself fixed by the condition
(2)m(T)+y(T)=K+n,
where n=m+y denotes the initial number of molecules in the compartment and *K* is a parameter that represents the number of new strands that can be created during the replication process, due to the finite amount of nutrients present in the compartment. We call it the carrying capacity. We use in the following the shorthands: $\bar{m}=m\left(T\right)$ and $\bar{n}=m\left(T\right)+y\left(T\right)$. Moreover, the ratio Λ=γ/α of the replicating constants of both species is another important parameter of the dynamics.

After the maturation step, the contents of the compartments are pooled. The fraction $x^{\prime }$ of replicases in the pool is expressed in terms of its value *x* at the beginning of the round by the following equation:(3)$$x^{\prime }\left(x,\lambda \right)=\frac{\langle \bar{m}\rangle }{\langle \bar{n}\rangle }=\frac{\sum_{n,m}\bar{m}\left(n,m\right)P_{\lambda }\left(n,m,x\right)}{\sum_{n,m}\bar{n}\left(n,m\right)P_{\lambda }\left(n,m,x\right)},$$
where 〈⋯〉 denotes the average with respect to the probability distribution $P_{\lambda }\left(n,m,x\right)$. Note that the number of molecules in the compartments at the end of the maturation step is not uniform (in particular, compartments which are pure in parasites contain at the end the same number of molecules as in the beginning). Thus, we cannot directly average *x* over the compartments as done in [10].

In Appendix B, we show that, in the limit Λ≫1 where the parasites are much more aggressive than the self-replicating molecules, the recursion equation (Equation 3) can be simplified, yielding
(4)$$x^{\prime }\left(\lambda ,x\right)=\frac{\lambda x+Ke^{-\lambda }\left(e^{\lambda x}-1\right)}{\lambda +K(1-e^{-\lambda x})}.$$

The behavior of this model is described in Section 3.1.

### 2.2. Transient Compartmentalization with Variable Inoculum Size

We now consider a model of transient compartmentalization with a variable inoculum size λ. In experiments based on serial transfers, a fraction of the solution is transferred into a new fresh medium repeatedly [17]. This can be described theoretically by adding a dilution step in the replicating cycle, as shown in Figure 1b. Then, λ can change because a given amount of the pooling solution can contain a variable number of replicating molecules, depending on their average concentration. The dynamics is now described by a pair of equations for the evolution of the fraction *x* and of the parameter λ:(5)$$x^{\prime }=\frac{\langle \bar{m}\rangle }{\langle \bar{n}\rangle }\;,\;\lambda^{\prime }=\frac{\langle \bar{n}\rangle }{d}\;.$$
where *d* is the dilution factor and $\bar{m}$, $\bar{n}$ are given by the same equations as above, evaluated with the current value of λ.

Using the same approximations used to derive Equation (Equation 4), we obtain the following set of equations, valid for Λ≫1:(6)$$\begin{array}{cc} x^{\prime }\left(\lambda ,x\right) & =\frac{\lambda x+Ke^{-\lambda }\left(e^{\lambda x}-1\right)}{\lambda +K(1-e^{-\lambda x})},\\ \lambda^{\prime }\left(\lambda ,x\right) & =\frac{\lambda +K(1-e^{-\lambda x})}{d}. \end{array}$$

These equations can be more easily manipulated than the corresponding equations for the global compartmentalization process.

The behavior of this model is described in Section 3.2.

## 3. Results

We first describe the results of the model with fixed inoculum size (Figure 1a, Section 2.1), and then those of the variable inoculum-size model (Figure 1b, Section 2.2).

### 3.1. Fixed Inoculum Size

By studying the stability of the fixed points of the recursion of Equation (Equation 4), we obtain the phase diagram shown in Figure 2, which represent the compositions that are accessible to the system on long times. In contrast to the phase diagram obtained in [10], we find a large region of coexistence between the self-replicating molecules and the parasites and no pure parasite phase. The absence of the pure parasite phase is expected, since parasites can not grow without replicators. Thus, there are only two phases: a pure replicator phase and a coexistence phase, in which compartments remain of mixed composition. Although the coexistence region appears large, in fact, in the main part of it, self-replicating molecules are maintained at a very small concentration, as shown by the color scale in Figure 2. Therefore, to maintain replicators at a significant concentration, one can not escape the condition that the average size of compartments be of the order of one molecule per compartment.

By evaluating the derivative of $x^{\prime }$ with respect to *x* at x=1, one obtains the equation of the vertical asymptote of the phase diagram separating the pure replicator phase and the coexistence region, which is given by λ=1. The same condition used at x=0 shows that this fixed point is always unstable at finite value of λ, which confirms that there is no pure parasite phase. In the coexistence region, a family of vertical asymptotes can be obtained by solving the equation $x^{\prime }\left(\lambda ,x\right)=x$ for 0<x<1 in terms of λ. No simple expression has been found for the equations of the corresponding horizontal asymptotes, which separate the pure replicator phase and the coexistence region when λ→∞.

### 3.2. Variable Inoculum Size

The most striking feature of the model with variable inoculum size, described by Equation (Equation 5), is the appearance of oscillations in both *x* and λ. They are similar to the ones observed in experiments with host-parasite RNAs [8] and modeled numerically in [16].

When the value of *K* and of *d* are not too large, this system exhibits oscillations in the populations of replicators and parasites shown in Figure 3a. These oscillations can also be seen when representing the fraction *x* of replicators as function of the average compartment size λ, as shown in Figure 3b. This behavior can be explained as follows: after the first inoculation, parasites are being replicated quickly by replicators, and the dilution does not counterbalance this increase in population. At some point, the fraction of parasites in the population is so high that there are not enough self-replicators to contribute to their replication. Then, the dilution has an important effect, since it decreases the population per compartment λ, until its average reaches values around λ=1 (see Figure 3b). At this point, compartments contain on average only a single molecule, which can be either a parasite or a self-replicator. The population in empty compartments or compartments containing a single parasite molecule does not grow, therefore only compartments containing a single replicator or containing one replicator and one parasite will contribute substantially to the next round. At this point, the replicator population increases, and starts replicating parasites for several rounds, triggering the process again. Note that the mechanism producing these oscillations is different from the Lotka–Volterra one, where the competition between the two species is the main ingredient [18]; instead, here, transient compartmentalization plays an essential role.

To delve deeper in the analysis of these oscillations, let us proceed with the equations in Equation (Equation 6), which are valid in the limit Λ≫1. In a simulation of these equations at a given value of *K*, we observe an abrupt transition when varying the dilution factor. Indeed, when K=60 and d=18, the two average populations oscillate steadily, as shown in Figure 4a, while when d=22, oscillations quickly die out as shown in Figure 4b. This abrupt transition is the sign of a bifurcation, which we identify as a supercritical Hopf bifurcation (see Appendix C for more details). The bifurcation occurs at d=20.74 given that K=60. Below this value, the system shows unstable spirals and converges to a limit cycle, while above this value, the system shows stable spirals which converge towards a fixed point (cf. [19], Section 8.2).

When the parameter *d* is further increased still keeping *K* fixed, we find a second transition at d=37.15. At this point, the system no longer oscillates or spirals around a fixed point, but instead converges towards this fixed point monotonically, a case identified as stable node in the literature ([19], p. 128).

Another interesting feature in these oscillations is the beating pattern which is visible on Figure 4a as a modulation in the amplitude of the oscillations. This pattern results from the interplay between two frequencies, the sampling frequency fixed by the duration of a single round, and the intrinsic frequency of the oscillations. By changing the sampling frequency, the beating pattern is accordingly modified.

To summarize all these results, we build the phase diagram in the plane (*K*, *d*) shown in Figure 5a. As can be seen in this figure, there are three phases in this diagram, which are separated by line boundaries in the plane (*K*, *d*). The upper region corresponds to a phase of unstable spirals [19], where steady oscillations are present; an intermediate region of stable spirals [19], where damped oscillations are present; and the lower region of stable node [19], where oscillations are absent. The presence of these line boundaries can be understood from the following argument. For large values of *K*, the equations which determine the fixed point coordinates ($x^{*}$, $\lambda^{*}$) deduced from Equation (Equation 6) can be simplified to yield:(7)$$\begin{array}{cc} x^{*} & =\frac{e^{-\lambda^{*}}\left(e^{\lambda^{*}x^{*}}-1\right)}{1-e^{-\lambda^{*}x^{*}}},\\ \lambda^{*} & =\frac{K(1-e^{-\lambda^{*}x^{*}})}{d}. \end{array}$$

The second equation above can be written as
(8)$$\frac{K}{d}=\frac{\lambda^{*}}{1-e^{-\lambda^{*}x^{*}}},$$
which shows that the coordinates of the fixed point ($x^{*}$, $\lambda^{*}$) only depend on the ratio K/d in this large *K* limit. It follows that the boundary between the region of unstable and stable spirals, where the Hopf bifurcation occurs is a straight line as shown in Figure 5a. A similar argument holds for the boundary between the stable spirals and the stable node, which is also a straight line in this diagram.

To confirm this interpretation, we show in Figure 5b the values of ($x^{*}$, $\lambda^{*}$) as a function of *K*, evaluated either on the boundary of the Hopf bifurcation and denoted with the subscript “hopf”, or on the stable node-stable spirals boundary and denoted with the subscript “osc”. When reporting the asymptotic values of ($x^{*}$, $\lambda^{*}$) obtained for large *K* in Equation (Equation 8), one recovers the values of the slopes of two lines in Figure 5a.

## 4. Discussion

We studied a simple system composed of a self-replicating molecule (a replicase) and a parasite molecule that needs the replicase for copying itself. In the case of a fixed inoculum size (i.e., for a fixed value of the parameter λ), we found that this system is able to maintain the replicase molecules against the take-over of parasites in the absence of artificial selection. Although the phase diagram contains a large coexistence region, only in a small part of it, when λ is close to one, are the replicase molecules maintained at a significant concentration. This may explain why experiments on directed evolution using compartmentalized self-replicating molecules such as DNA or RNA are usually carried out in this regime for these molecules, while all other required chemical species (nucleotides, other intermediates, etc.) are typically in excess.

The theoretical framework we developed for the case of a variable inoculum size has many similarities with the model proposed in [16] to explain experiments on host-parasite RNAs [8]. There are however some differences: we consider an infinite number of compartments instead of a finite one, we do not include mutations which could turn the replicase into a parasite, and we do not include local mixing, which means that our model corresponds to the infinite mixing limit of ref [16]. Despite these differences, we also find a regime of values of the parameters (in particular for the dilution factor or the carrying capacity) in which sustained oscillations are possible in agreement with Furubayashi and Ichihashi [16]. In [8], the ratio of catalysis rate constants of the parasite with respect to host, which we denote Λ, was about 5: this can be obtained by extracting, from Table I in that reference, values of α=0.29,γ=1.5 and using Λ=γ/α≃5. We find that oscillations are indeed present in our model in this range of the parameters, and oscillation periods comparable to the value reported in [8] can be recovered from this estimate. These similarities suggest that our assumptions of an infinite number of compartments may be reasonable and that other differences related to local mixing and pooling may not be essential. To test these points more precisely, it would be useful to redo experiments similar to the ones of Bansho et al. [8] but with a pooling step done as in [7].

It appears that, for both fixed and variable inoculum sizes, the regime of pure compartments (one molecule per compartment on average) has a particular importance: for a fixed inoculum size, only in this regime can a significant average fraction of self-replicating molecules be maintained, and for a variable inoculum size, only in this regime can a rebound occur in the populations of molecules, allowing oscillations. We surmise that this regime could have a specific significance for the origin of life. To elaborate a bit on this point, we recall that the emergence of special molecules bearing the genetic information is an essential step in the origin of life as emphasized in the RNA world. These molecules are typically found in minority with respect to other species, yet this minority has control of the entire cell [20]. This is a form of information control, which is thought to be one of the key parameters in the origins of life [21]. Fluctuations of this minority species therefore have a special role due to their small number. In contrast, many other chemical species, which are not information carriers, are found in large numbers, with fluctuations statistically obeying the law of large numbers. In our model, we see a clear illustration of this mechanism: the replicase behaves as a genome-like molecule, present at the lowest non-zero possible concentration of one molecule per compartment, while all other molecules, which depend on the genome molecules for their own making, are available in the protocell in large concentration.

## 5. Conclusions

Without considering complex chemistry, we have proposed a model which is able to capture important features for origins of life research, such as the ability to maintain self-replicating molecules using transient compartmentalization and natural selection. An interesting feature of the model with constant inoculum size is the maintenance of the self-replicating molecule by a form of information control, at the critical level of one molecule per compartment. A striking feature of the model with variable inoculum size is the appearance of oscillations, which are similar to the ones observed in experiments with compartmentalized self-replicating RNAs [8].

Naturally, Bansho et al. [8] presented much more than the mere observation of these oscillations. By studying the sequence information of the replicase and its parasites, they suggested that parasites can take an active part in the evolution of their host and not just in their own. Different sub-populations of parasites can appear, forming an ecosystem [22], which accelerates evolution. Future studies are needed to quantify these co-evolutionary mechanisms, and perhaps our model could help in that task.

Another important direction for future work would be to consider a large number of interacting chemical species, a situation frequently encountered in statistical physics [23]. In this case, we expect that the basic unit of description may no longer be that of single chemical species, but could become collective excitations of the composition, similar to quasi-species [24] or composomes [25]. A general theory of non-equilibrium chemical networks, constrained by conservation laws and symmetries, has recently been put forward [26,27]. One attractive feature of such a framework for describing complex chemical systems is that it relies mainly on stoichiometry; therefore, the explicit knowledge of the kinetics, which is often missing, is not needed [17].

New types of emergent behaviors could arise by enlarging further the dynamics of compartmentalization. One possibility would be to consider loose compartments [28] or a continuous automated in vitro evolution [29], which will require introducing spatially dependent parameters in our transient compartmentalization dynamics. Besides the relevance for the origins of life, we hope that our work could trigger new research directions on applications of transient compartmentalization for chemistry or biochemistry. Perhaps, these new research directions could help overcome practical and fundamental hurdles associated with the synthesis of complex molecules, and facilitate the making of new catalysts or artificial cells [30].

## Figures and Tables

**Figure 1 life-09-00078-f001:**
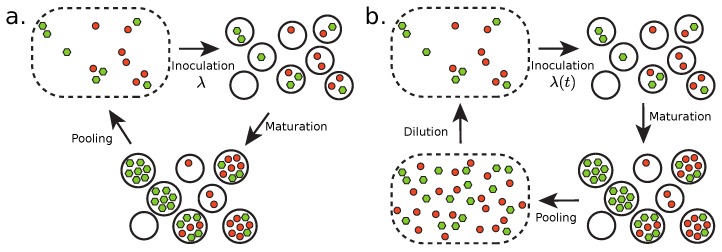
(**a**) Transient compartmentalization at fixed average number of molecules per compartment, and (**b**) with a variable average number of molecules. In (**a**), the cycle splits into steps of inoculation, with a fixed average number of molecules per compartment λ, maturation and then pooling, while in (**b**) the inoculation step is done with a variable average number of molecules per compartment λ(t) because the cycle contains in addition a dilution step. The green and red circles represent the replicators and their parasites, respectively.

**Figure 2 life-09-00078-f002:**
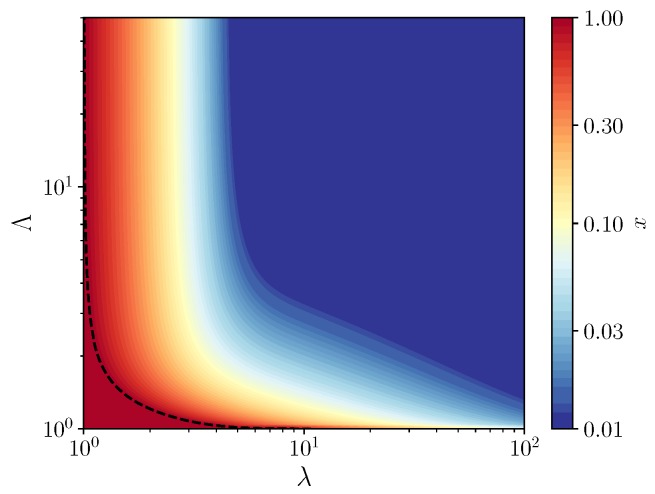
Contour map of the fraction *x* of replicators as a function of (λ,Λ), for a carrying capacity K=100, where λ denotes the average number of molecules per compartment and Λ the relative growth rates of the parasites with respect to the host. The dotted line is the contour of x=1, which marks the border of the pure replicators phase (the red region). Above this line, a coexistence region exists between the two species at a fraction of replicators indicated by the color scale.

**Figure 3 life-09-00078-f003:**
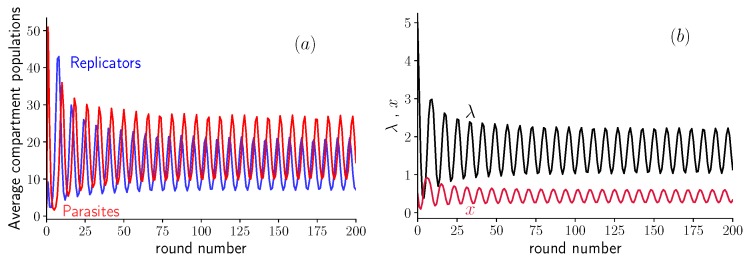
Oscillations in the average amount of self-replicating and parasite molecules per compartment as a function of the round number for d=19, K=60, and Λ=5. (**a**) Average population size $\langle \bar{m}\rangle$ of replicators and $\langle \bar{n}-\bar{m}\rangle$ of parasites after the growth step plotted vs. round number. (**b**) Fraction *x* of replicators and average λ of inoculum size. Notice that the oscillations rebound close to the line λ=1.

**Figure 4 life-09-00078-f004:**
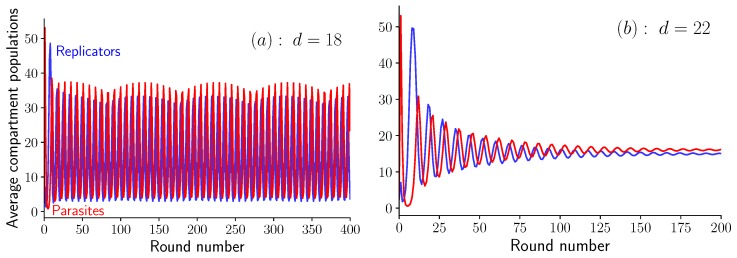
Oscillations in the average amount of self-replicating and parasites molecules per compartment as a function of the round number for K=60 and Λ≫1. (**a**) Steady oscillations at d=18 (unstable spirals), and (**b**) damped oscillations at d=22 (stable spirals). Note the beating pattern in the oscillations visible in (**a**).

**Figure 5 life-09-00078-f005:**
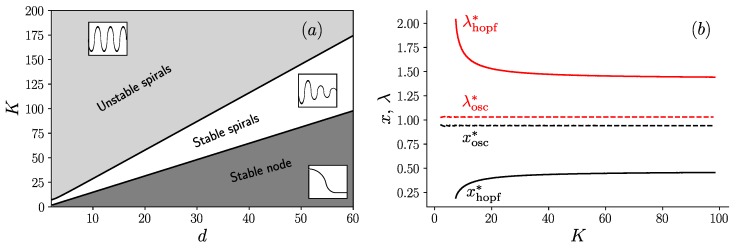
(**a**) Phase diagram in the plane (*K*, *d*) in the limit Λ≫1 containing three regions: unstable spirals (with an inset representing steady oscillations), stable spirals (with an inset representing damped oscillations) and stable node (with an inset representing a curve with no oscillations); and (**b**) evolution of the fixed point coordinates ($x^{*}$, $\lambda^{*}$) as a function of *K*, on the Hopf bifurcation (solid line) and on the transition line between the stable node and stable spirals (dashed line).

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
