# Peer review of "Survival of Self-Replicating Molecules under Transient Compartmentalization with Natural Selection"

_life, 2019, doi:10.3390/life9040078_

Round 1

Reviewer 1 Report

The manuscript by Laurent et al, entitled “Survival of self-replicating molecules under transient compartmentalization with natural selection”, address the effect of transient compartmentalization and dynamics of a possible ancient replicase and parasites. They analyzed the population dynamics both analytically and numerically and provide a rigid theoretical framework for the understanding of the importance of transient compartments for the origin of life studies. The manuscript is well-organized and precise. I believe that this study is very important for the origin of life community. I recommend publishing this work after addressing the following minor points.

Page 7. In the sentence that starts from 3rd lane (without line numbers), the author wrote: “In the limit K -> infinity, the equations which determine the fixed……”. The statement, “In the limit K -> infinity”, seems strange for me because K remains in equation 7. How about to use just “large K condition” or equivalent words?

The last sentence in the Result, “When either K or d is very large, one does not expect any oscillation as shown in Fig. 5a, was difficult to understand because with a large K, the dynamics are in Unstable spirals according to Fig. 5a and in Unstable spiral phase, the replicases and parasites should oscillate as described in Fig. 4a.

Reviewer 2 Report

This is a very nice paper that presents a simple and elegant model dealing with the infamous error catastrophe in the context of the origins of life.  The central message is that compartmentalization allows suppressing the behavior when a system of replicators ends up producing mostly parasitic molecules rather than themselves. For this, the size of the compartment should not be fixed, leading to a striking oscillatory behavior of the overall composition. 

I have a number of suggestions on how the manuscript could be improved before being accepted for publication. 

1)  The part of the manuscript where the connection to prior experiment, Ref 15 and model,  Ref 16, is discussed, is very sketchy. It reads as if it is included primarily to address a possible criticism from the authors of those papers.  Instead, the context should be properly introduced to an unprepared reader, after which the specific difference between the present work and those papers could be discussed. 

 2) The paper would benefit if a specific design of a possible experimental test were suggested. 

3) The model does not include any spatial variability. It is, however, an important question,  whether the behavior would survive if one allowed the composition, size of the compartment and phase/frequency of the oscillations to vary in space. Can the authors comment on this? 

Reviewer 3 Report

In this theoretical work the authors study the behavior of a self-replicating molecule in the presence of a parasite molecule when transient compartmentalization takes place. Two different scenarios are addressed:
with fixed sized compartments and with variable inoculum size. The former case results in stationary states, while the latter allows the formation of oscillatory states. During their analysis they also consider the limit where parasite molecules are more aggressive in the replication process.
In the paper the governing equations are well defined, the analysis is correctly presented. The details are placed in the appendix, where they provide useful information without interrupting the main flow of the text. The work demonstrates an aspect of transient compartmentalization that may be important in relation to the origin of life.

I recommend the publication of the manuscript after the authors have considered the following change:
Figure 1 does not reflect the difference between the two models studied: the fixed and variable inoculum size. A modified figure would help the readers in capturing the key difference.
